# Tracheal Tissue Engineering: Principles and State of the Art

**DOI:** 10.3390/bioengineering11020198

**Published:** 2024-02-19

**Authors:** Marco Mammana, Alessandro Bonis, Vincenzo Verzeletti, Andrea Dell’Amore, Federico Rea

**Affiliations:** Department of Cardiac, Thoracic and Vascular Sciences, University of Padua, 35128 Padua, Italy; alessandro.bonis@aopd.veneto.it (A.B.); vverzeletti@gmail.com (V.V.);

**Keywords:** trachea, tissue engineering, transplantation, stem cells

## Abstract

Patients affected by long-segment tracheal defects or stenoses represent an unsolved surgical issue, since they cannot be treated with the conventional surgery of tracheal resection and consequent anastomosis. Hence, different strategies for tracheal replacement have been proposed (synthetic materials, aortic allografts, transplantation, autologous tissue composites, and tissue engineering), each with advantages and drawbacks. Tracheal tissue engineering, on the other hand, aims at recreating a fully functional tracheal substitute, without the need for the patient to receive lifelong immunosuppression or endotracheal stents. Tissue engineering approaches involve the use of a scaffold, stem cells, and humoral signals. This paper reviews the main aspects of tracheal TE, starting from the choice of the scaffold to the type of stem cells that can be used to seed the scaffold, the methods for their culture and expansion, the issue of graft revascularization at the moment of in vivo implantation, and experimental models of tracheal research. Moreover, a critical insight on the state of the art of tracheal tissue engineering is also presented.

## 1. Introduction

Severe tracheal stenosis, tracheomalacia, tracheoesophageal fistula, and primary and secondary tracheal tumors all represent potential indications for resection of the affected airway segment and subsequent end-to-end anastomosis. However, defects or stenoses involving more than half of the tracheal length in adults and more than one-third in children represent an unsolved surgical issue; in fact, these defects cannot be repaired by simply reapproximating the healthy tracheal edges because this would result in excessive tension on the anastomosis and in potentially life-threatening complications. Unfortunately, the only option for these patients is represented by long-term airway stenting with endotracheal stents. However, these devices present some drawbacks, such as the need to be changed regularly, the possibility of migration, granulation tissue formation, stenosis, hemorrhage, and fistula, with complications that occur in a not insignificant percentage of patients [1]. In order to address this unmet clinical need, different strategies for tracheal replacement have been proposed, including the use of synthetic materials, aortic or tracheal allografts, tracheal transplantation, autologous tissue composites, and tracheal engineering [2]. Some of these experimental procedures have successfully been performed on humans with satisfactory results; however, none of them is devoid of important drawbacks [3].

Synthetic materials were the first to be investigated for tracheal replacement; however, clinical experience with the use of either solid or porous prostheses has been unsatisfactory. In fact, poor biocompatibility, death, and anastomotic complications have been repeatedly reported in early case series, and this replacement option now seems to have been abandoned [4,5].

Tracheal transplantation, on the other hand, has been carried out successfully by different authors [6,7].

However, the need for immunosuppression represents an absolute contraindication in malignant stenoses and also poses an ethical concern for patients with benign stenoses, who can be managed safely with long-term airway stenting [8]. Similarly, airway replacement with aortic allografts proved successful in a single-center experience of 35 patients, some of whom were affected by primary tracheal neoplasms; but only 28.6% of enrolled subjects achieved stent-free survival [9].

Tissue engineering (TE) allows for the in vitro or in vivo creation of tracheal tissues by associating a three-dimensional (3D) scaffold with the patient’s stem cells [2]. Compared with the other treatment options mentioned above, there is much less clinical evidence about the feasibility of TE approaches for tracheal replacement. Indeed, anecdotal attempts have been made to transplant a tissue-engineered trachea in humans, as a compassionate treatment, with conflicting results [10].

On the other hand, there are many expectations of TE approaches, since, contrary to other options, they have the potential to create an ideal tracheal substitute. In fact, TE approaches aim to create a non-immunogenic construct by the use of the patient’s own stem cells with ideal mechanical properties and with a fully functional ciliary respiratory epithelium [11]. Given these premises and the obvious clinical impact that this technique could have, it is not difficult to understand why papers about this fascinating topic are appearing more and more frequently in the recent literature.

The aim of this paper is to review preclinical evidence concerning the use of stem cell-based tissue engineering approaches for tracheal replacement in order to gain insight on available evidence and to better understand the hurdles that still need to be overcome before TE approaches become a reality for patients affected by long-segment tracheal defects.

## 2. Anatomy and Physiology of the Trachea

The trachea is a hollow, tubular structure; its length normally ranges between 10 and 14 cm [12,13]. It extends cranio-caudally from the sixth cervical vertebra to the fourth thoracic vertebra, where it divides into the two main bronchi [12,13]. It is characterized by a typical D-section structure, supported anteriorly by 16–20 incomplete rings of hyaline cartilage (pars cartilaginea) connected to each other by fibroelastic tissue. The posterior wall is flattened and is formed by a muscular structure (pars membranacea) [12,13]. Each tracheal ring has an average height of 4 mm. The outer diameter is about 2.2–2.0 cm in men and 2.0–1.8 cm in women [12,13].

The trachea presents segmental vascularization, which means that the arteries first enter the organ wall laterally, then anastomose longitudinally with the superior and inferior segmental arteries [12,13,14]. Furthermore, each segmental artery divides into an anterior and a posterior branch that encircle the trachea and anastomoses with the contralateral segmental artery [12,13,14]. The main arterial vessels from which this segmental network arises are, from the right side, the bronchial artery and the inferior thyroid artery; while from the left side the left side, the main arteries are the middle and inferior thyroid arteries and the internal thoracic artery [12,13]. 

Regarding tracheal architecture, it consists of four different layers: the mucosa and submucosa, common to the whole organ, and a deeper layer represented by the cartilaginous scaffold on the antero-lateral side and by the membranous layer posteriorly [12,13]. The outermost layer is the tunica adventitia [12,13]. 

The tracheal luminal mucosa is lined by a pseudostratified columnar epithelium characterized by ciliated, brush, basal, and secretory cells (i.e., globular cells) on a basal membrane populated by epithelial cells [12,13]. The basic function of the airway epithelium is to provide a barrier against respiratory pathogens [10,12,13].

In the submucosa, on the other hand, globular cells and mucous glands provide for the secretion of a protective layer of mucus [10,12,13]. Mucus guarantees the fundamental function of the trachea. By keeping the surfaces of the tracheal epithelium moist and lubricated, mucus has the function of capturing particles, microorganisms, mucosal flaking cells, and leukocytes present during inflammatory reactions, facilitating their clearance.

Finally, the tracheal cartilage extracellular matrix is characterized by collagen fibers entrapping a matrix core of hydrated proteoglycans [10,12,13]. This composition underlies the biomechanical characteristics of the trachea that an efficient substitute should ensure. It needs to accommodate the tensile forces surrounding trachea, and it must present stiffness and rigidity on lateral walls, as well as longitudinal flexibility. In particular, rigidity and flexibility are extremely relevant: in fact, they combine to avoid collapse of the tracheal structure during changes in airway pressure related to respiratory acts [10,12,13].

## 3. Approaches to Tracheal Tissue Engineering

As mentioned above, tissue engineering consists in creating a tracheal substitute by populating a scaffold with stem cells. Therefore, the process of tissue engineering involves several key components, including scaffolds, cells, and vascularization (Table 1). As proposed by Fishman and colleagues, a first major distinction can be made based on the modality and timing employed to let the stem cells interact with the scaffold [15].

In vitro TE consists of seeding stem cells onto the scaffold in the laboratory prior to in vivo implantation. Stem cells are then cultured onto the scaffold either statically or by the use of a bioreactor. In vivo implantation occurs after a variable time, ideally when the scaffold has been sufficiently populated by stem cells to become fully functional. In in vivo TE, on the other hand, stem cells are seeded onto the scaffold directly at the time of in vivo implantation, using the recipient’s body as a “living bioreactor”. Lastly, in situ TE consists of the implantation of the scaffold alone, relying on the recruitment and homing capabilities of endogenous stem cells from blood vessels and surrounding structures [15].

The majority of published studies on the use of stem cells for tracheal replacement describes an in vitro TE approach, wherein stem cells are incubated with the scaffold for a variable amount of time prior to implantation, with few exceptions [16,17].

In a study by Haykal et al., recipients’ mesenchymal stem cells (MSCs) were seeded over the external surface of the scaffold (decellularized trachea) at the moment of heterotopic implantation under the sternocleidomastoid muscle, just prior to final suturing of the muscle flap around the tracheal segment. However, in this study, MSCs were used mainly to study their immunomodulating properties rather than their capabilities to repopulate the scaffold [16]. In a study by Wood et al., on the other hand, decellularized tracheal scaffolds were orthotopically transplanted in a canine model. The day of surgery, recipient’ adipose-derived MSCs were suspended in fibrin glue and applied topically to the graft’s lumen around a separate, sterile endotracheal tube to allow for circumferential coating while maintaining the patency of the lumen [17].

Unfortunately, all animals died between the 9th and 11th postoperative days, and no specific test could be conducted to test the success of MSC seeding.

## 4. Scaffold Options

The scaffold is a key component for the success of the TE approach. Ideally, the material chosen as a scaffold should be biocompatible, non-immunogenic, and supportive of three-dimensional cell growth, with appropriate microarchitecture and functional properties, adequate mechanical properties, bioactive and chemoattractant properties, controllable degradation, and porosity. Furthermore, from the practical point of view, the ideal scaffold should be readily available and inexpensive [15].

Scaffolds can be divided into natural and synthetic ones. Both material types have advantages and drawbacks. Natural scaffolds are derived from donor tissues (e.g., donor trachea) and are treated by decellularization to remove any immunogenic cellular components. Decellularization can be achieved by a combination of chemical, physical, or enzymatic strategies. The aim of an ideal decellularization protocol is to remove all cellular components while, at the same time, preserving the extracellular matrix composition and angiogenic factors. Unfortunately, there is still no agreement to date as to which is the optimal decellularization protocol for tracheal tissue. Moreover, several aspects related to decellularization, such as the number of decellularization cycles and the “grade” of decellularization, are still under debate. In fact, a completely decellularized scaffold may have inferior mechanical properties relative to a partially decellularized scaffold, leading to collapse or stenosis after implantation. Therefore, the thoroughness of decellularization may not guarantee better biocompatibility. Unfortunately, well-designed preclinical studies that compare different decellularization grades are lacking [10].

The advantages of a natural scaffold include good biocompatibility, as well as the presence of an extracellular matrix (ECM) that possesses the same native tissue composition, microarchitecture, microvasculature, and signaling components. On the other hand, decellularized tissues, once implanted, inevitably undergo resorption and degradation; moreover, because of both the decellularization process and tissue degradation after implantation, the rigidity of the scaffold is often not adequate to maintain a patent airway lumen. Lastly, a natural scaffold requires an animal donor source, which may be limited, and a variable time for the decellularization process, which usually takes weeks to months.

Synthetic scaffolds overcome many of the aforementioned practical limitations, since they can be available off-the-shelf, and they do not suffer from restrictions due to donor tissue availability. These materials are usually designed to have optimal mechanical properties, and their resorption rate, once implanted, can be controlled. On the other hand, they may be less biocompatible than natural scaffolds, and their microarchitecture does not resemble native tissues. In a study conducted by Pepper et al. on a sheep model, for instance, the authors highlighted several issues related to artificial prostheses [18]. First of all, they experienced a high incidence of infectious complications (four of eight had pneumonia, three of eight had lung abscess, and all animals presented at least fibrinopurulent exudate); secondarily, the prostheses caused mechanical complications (posterior wall infolding, scaffold delamination, and graft dislodgement) and inflammatory aberrancies (stenosis or granulation), which resulted in an insufficient formation of the neo-epithelium. They speculated that this could be related to a lack of epithelization [18]. Similar findings resulted from a study conducted by Fux et al. on three human patients who received synthetic tracheal grafts seeded with bone-marrow cells. All patients experienced graft-related complications, necessitating multiple surgical reinterventions. Even after a long follow-up period, there were no signs of graft vascularization, mucosal lining, or integration into adjacent tissues [19]. According to the scarce literature available on humans, the experience related to synthetic prostheses has been largely unsatisfactory, being characterized by postoperative early and late major complications (suture granulomas or dehiscence, fistulas, infectious complications, and death) [20,21,22].

Over time, a wide variety of resorbable materials has been tested with the aim of mimicking native tissues as much as possible; among them, the most common are polycaprolactone (PCL) and polyurethane (PU). These two polymers can be combined in order to modulate the elastic properties and the degradation rate of the scaffold [23].

Better results in terms of biocompatibility, mechanical properties, and replacement feasibility have been introduced with hydrogels and gelatin-derived scaffolds, as they are more adherent to the ECM structure and properties. 

Gelatin is obtained by hydrolysis of collagen, a widely represented ECM protein in various ranges of applications, from cell seeding and cultures in preclinical research to tissue replacement [24]. Despite its biocompatibility and the low cost, gelatin leads to scarce mechanical properties, a high enzymatic degradation rate, and reduced solubility in liquid media [25]. A mixed gelatin–saccharide structure was introduced in an attempt to overcome those limitations by reproducing an ECM-like environment and increasing the water incorporation ability. Once gelatin is crosslinked, a form of hydrogel is obtained. Biocompatibility and low antigenicity are the main characteristics that have elicited engineering research on gelatin, while lower physical properties have led to a preference for hydrogels [26].

Hydrogel are flexible, elastic, and thick crosslinked polymers able to retain water without dissolving in it. They can be natural (fibrin, hyaluronic acid, or alginate), semisynthetic (PEG–alginate) or synthetic (PEG or PVA). The great biocompatibility of hydrogels, which is maintained in semisynthetic and synthetic structures, as well as their versatility, has led to their widespread utilization in recent years in several research fields, such as tissue engineering and in precision medicine [27,28].

In particular, regenerative hydrogels have been adopted in regenerative medicine to encourage cell growth and sustain a viable microenvironment [29]. In the field of tracheal bioengineering, the structure of hydrogels (3D multilayer polymer) has been proven to be a valid surrogate for the extracellular matrix (ECM), sustaining cells’ organization, viability, and growth [30] and avoiding bacterial cell growth [31,32]. In addition to biocompatibility, the similarities with chondrocyte-derived tissues (such as cartilage) constitute a further argument for their utilization [33], and this matrix is extensively adopted in tracheal tissue engineering [34,35].

Besides the choice of the material, the manufacturing method has its own importance. Most common methods include molding, electrospinning, and 3D printing.

The first method is probably the most intuitive and least complex; scaffolds are created from preformed molds—which can be designed based on the dimensions of the tracheal defect to be replaced—into which the material is then cast.

The electrospinning technique exploits the electric field generated by two different electrodes for polymer deposition to produce porous substrates of ultra-fine fibers with a large surface area, which makes them ideal environments for cell growth and new tissue formation [36]. Finally, the third method involves the creation of the scaffold from a virtual render, from which the 3D model is obtained and printed. Usually, particularly referring to preclinical animal studies and for the few attempts on humans, such a render is obtained from a semiautomatic segmentation process based on the defect that the scaffold should replace. Segmentation is performed with dedicated software starting from a radiological study (usually a CT scan) up to the creation of an .*stl* file that is then modified according to the needs of the different 3D printers. The above process leads to the formation of extremely accurate scaffolds with discrepancies from the virtual render of less than 1 mm [37].

Concerning 3D-printed models, the literature reports several experiences in which three-dimensional upper airways substitutes were created with the aim of replacing circumferential or segmental tracheal defects [38,39,40]. In particular, a new bioengineering field called 3D bioprinting is quickly developing. It is a process that consists of creating a layer-by-layer structure made of natural or synthetic materials and living cells. It is divided into three main phases: first, the size of defect that needs to be managed (mini-plates, splints, or non-circumferential or circumferential defects); second, the scaffold material (PGA, PLA, PCL, or PLCA); and, finally, the cell-containing hydrogel (such as alginate or gelatin) [41]. Printed models present several shortcomings and aspects that need to be detailed. First of all, the macroscopic shape of the model is as fundamental as its microscopic and nanoscopic shape due to biological processes that surround positive graft integration (such as cell migration and proliferation and ECM remodeling). This is one of the leading open problems as concerns 3D-bioprinted models: to obtain a defect replacement with competitive mechanical and physical properties (macroscopic shape) and, at the same time, to deliver a micro- and nanoscopic structure able to induce cell migration, complete proliferation and surface coverage, and activate living cells. From this perspective, advances in terms of computed modeling and printing are going to delineate future directions [42].

Freeze drying (or lyophilization) is a relatively older technique that uses cold temperatures to obtain deep dehydration by managing pressure and temperature and, finally, inducing ice sublimation [43]. In tracheal engineering, the obtained scaffold is a dehydrated, decellularized trachea. This process may be associated with sonication to improve decellularization, which is not always sufficient with freeze drying only [44].

Recently, Khalid et al. produced a reinforced collagen–hyaluronic acid tracheal scaffold by a combined method involving the use of a 3D-printed PCL framework (obtained by Python programming) and subsequent reinforcement with the freeze-drying process [45].

Salt leaching is another method to develop three-dimensional matrices, with the possibility of controlling the porous size. The obtainable shape is extremely variable, while the structure remains uniform and easily modulable. It consists of a mixed solution (NaCl crystal added to a PCLA solution) molded in silicon preprinted shapes that are subsequently leached into a solution to exclude NaCl, maintaining only the PCLA component and shaped as desired [46]. This is extremely important due to the tracheal regenerative epithelialization process because it was previously suggested that pores size is related to tissue [47]. Compared to printed scaffolds, those created by salt leaching suffer from an inferior replicability and faster in vivo degradation due to their irregular structure [46].

Each of these manufacturing methods has inherent advantages and drawbacks, which are beyond the scope of this paper but are extensively covered elsewhere [36,48].

Lastly, a scaffold may be composed of multiple layers, where natural and synthetic tissues can be combined in order to overcome their respective limitations; these are referred to as hybrid scaffolds. An example is provided by the study of Ghorbani et al., who fabricated a scaffold whose outer layer was constituted by electrospun nanofibers composed of a blend of collagen and polycaprolactone (PCL), while the inner layer was made of decellularized aorta [23]. By doing so, the authors achieved a TE construct whose inner layer was biocompatible in order to avoid the formation of granulation tissue and favor epithelial cell ingrowth, while the outer layer had optimal mechanical and degradation properties. This scaffold was seeded in vitro with MSC for 21 days, showing rising mRNA levels of collagen I, collagen II, and aggrecan [49].

## 5. Cell Types for Tracheal Regeneration

Regardless of the material chosen as a scaffold, cell seeding is an essential component of a TE approach. In fact, while several attempts at transplantation of unseeded scaffolds have been reported (often as a comparison with seeded scaffolds), there is consistent evidence that these scaffolds are more immunogenic [16] and more likely to develop critical strictures, granulations, and infection, leading to early animal death whenever orthotopic transplantation is attempted [50,51]. There is no consensus in the literature as to what cell types should be seeded onto the scaffold. Besides stem cells, differentiated cell types are often used in preclinical studies. These include chondrocytes and tracheal epithelial cells. While MSCs can differentiate into chondrocytes with appropriate growth signals, differentiated chondrocytes can be successfully harvested and expanded from the auricular cartilage, the nasal septum, or the trachea [52]. Tracheal epithelial cells, on the other hand, are usually obtained from tracheal biopsies. According to a study by Go et al., both cell types seem necessary for survival of the implant. In their study, they demonstrated that pigs who were orthotopically transplanted decellularized matrices seeded with both chondrocytes on the outside and epithelial cells on the inside were the only animals who achieved survival until 60 days, with no signs of stenosis or bacterial contamination, compared with matrices seeded with either chondrocytes or epithelial cells alone [53].

The importance of chondrocytes is intuitive, as they contribute to the deposition of extracellular matrix protein and areas of “neocartilage” formation. The mechanisms by which epithelial cells are believed to contribute to the success of a TE approach are less clear.

Villalba-Caloca and colleagues proposed that early recellularization with chondrocytes leads to an improved structural and mechanical advantages, with a reduction in suture-related fibrosis [54]. Moreover, previous studies have demonstrated that seeding of chondrocytes into the inner graft’s layer could pivot the re-epithelization process [39,55]. Consequently, chondrocytes may not be merely considered a structural part of the trachea but as active participants in mechanical and biological graft properties.

Indeed, direct engraftment and regeneration of the epithelium by the seeded cells are unlikely to happen over the short term. However, these cells have the potential to act as a biological dressing, acting as a barrier against infections and stimulating regeneration from surrounding host cells [56].

## 6. The Role of Stem Cells

Stem cells have unique properties of self-renewal and multi-lineage differentiation, which are thought to be essential for tissue regeneration. Moreover, they possess potent immunosuppressive properties and paracrine activity [56]. Theoretically, different types of stem cells can be used, including embryonic stem cells, amniotic fluid-derived stem cells, induced pluripotent stem cells (iPSCs), and adult MSCs. For the purpose of tracheal TE, however, the vast majority of published studies involve the use of adult MSCs, either adipose-derived or bone marrow derived. These cells can be easily isolated and expanded in vitro and have well-characterized, multi-lineage differentiation capacity and paracrine activity [57,58]. Moreover, it has been demonstrated that bone marrow-derived MSCs share a substantial transcriptional profile with human lung MSCs, making them suitable for tracheal regeneration purposes [59].

An alternative source of MSCs is represented by the human umbilical cord. In a study by Baggio Simeoni et al., human umbilical cord MSCs were successfully differentiated into chondrocytes in vitro and were then incubated for 7 days with an acellular amniotic membrane. The resulting TE construct was implanted on a tracheal injury model of New Zealand rabbits. Histopathological analyses conducted 60 days after surgery revealed the presence of immature cartilage islands [60].

A study by Kim et al. is the only report investigating the use of human iPSCs for tracheal TE [61]. The authors fabricated a two-layered PCL scaffold through electrospinning (inner layer) and 3D printing (outer layer). The scaffold was seeded with human bronchial epithelial cells, iPSC-derived MSCs, and iPSC-derived chondrocytes, then cultivated in a bioreactor system for 2 days. The authors used their TE airway to repair a circumferential 1.5 cm tracheal defect in a rabbit and demonstrated that, after 4 weeks, all animals survived with maintained airway patency.

As mentioned above, MSCs have the potential to differentiate into various cell types, including chondrocytes. Therefore, it is unclear whether cultivating MSCs in a differentiation medium prior to seeding on the scaffold is a required step, as investigators have adopted different approaches in this regard. In fact, spontaneous chondrogenic differentiation of MSCs is often expected to occur as a result of the good biocompatibility of the scaffold itself [49]. In their study, Bae et al. fabricated a 3D-bioprinted, multilayered PCL scaffold, which was seeded with epithelial cells and MSCs, and studied the effect of culturing MSCs on a chondrogenic differentiation medium or a regular medium prior to seeding them on the scaffold. The authors observed that after in vivo implantation, scaffolds seeded with chondrogenic differentiated MSCs had the greatest potential for tracheal cartilage regeneration [62]. Overall, available evidence suggests that seeding a scaffold with different types of cells (usually MSCs and epithelial cells) is necessary in order to obtain an optimized tracheal substitute. This has been independently demonstrated by several studies [16,53,63]. MSCs, in particular, seem to have a pivotal role in co-seeding strategies, since they support appropriate proliferation and differentiation of other cell types.

Besides the chosen cell type and its differentiation, additional factors that deserve consideration are the density of seeded cells, which, in reported studies, varies from 1.6 × 10^5^ [60,63] to 1.5 × 10^7^ [62], and the efficiency of seeding, which varies according to the method used. Currently, there is no evidence available to define the minimum number of cells that should be retained in the scaffold before it can be successfully implanted. In a previous in vivo study, it was reported that the number of cells seeded was significantly related to the time to respiratory distress onset after synthetic tracheal replacement, suggesting a positive “dose–effect” relationship of stem cell seeding [18]. Moreover, it remains to be elucidated whether the beneficial role of MSCs, as well as that of other cell types, consists more in their permanent engraftment and retention rather than in their release of immunomodulatory, chemotactic, and trophic factors.

## 7. Seeding Methods and Bioreactors

The main distinction between seeding methods consists of static and dynamic methods. Static methods consist basically of incubating the scaffold with the cell suspension for a variable amount of time—typically, from 84 h to 1 week [49,64]. Dynamic methods, on the other hand, aim to recreate the microenvironment to which the trachea would be exposed in vivo by the use of a bioreactor. In general, a bioreactor allows for the culture of a greater volume of cells in vitro compared to static methods; moreover, a wide array of parameters can be monitored and optimized, including rotational speed and flow, as well as the composition of the perfusate. Bioreactors may be constituted by a rotating chamber [61] or a spinner flask [65] or have a more complex design [66]. More sophisticated systems allow for the differential control of the flow of culture medium and stem cells between the outer and the inner part of the tubular TE construct [66,67].

Lastly, scaffold cell repopulation may be favored by the use of growth factors, such as vascular endothelial growth factor (VEGF) [68] or transforming growth factor-β (TGF-β) [63,65,69].

TGF-β, in particular, has been demonstrated to improve the chondrogenic capacity of MSCs [63,65,69] or differentiation towards smooth muscle cells [63]. Taken together, the combination of scaffold, seeded cells, and humoral signals constitutes the classic tissue engineering paradigm. Optimization of all three components is necessary in order to generate optimized TE tracheal substitutes [70].

## 8. Vascular Supply

Regardless of the scaffold material, the type of seeded cells, and methods for their culture and expansion, it is unlikely that such cells will survive after in vivo implantation if an adequate blood supply is not ensured. The trachea is vascularized by a finely segmental blood supply without a dominant vascular pedicle, which makes it unsuitable for direct anastomosis. The vessels from which these branches arise derive from different vascular territories and are the external carotid artery and branches of the subclavian artery. The issue of graft vascularization is often overlooked in experimental studies of TE tracheal replacement that involve in vivo graft implantation models; however, there is evidence from tracheal transplantation studies that revascularization of an allogeneic trachea after in vivo implantation takes several weeks to months occur when wrapped into a well-vascularized recipient tissue [14].

Heterotopic revascularization in humans has been previously tested by wrapping the tracheal transplant in different tissues (i.e., the omentum, the soft tissues of the forearm, and the sterno-cleido-mastoid muscle) [7,71,72]. In particular, Delaere et al. have extensively studied—both in animal models and in humans—the physiology of heterotopic revascularization of an allogeneic tracheal graft under the protection of immunosuppression prior to orthotopic implantation. In their studies, revascularization of the allogeneic graft occurred within 2 weeks from heterotopic implantation in a rabbit model and within 2 to 3 months in humans [14,73]. In their first patient, before revascularization occurred, the posterior membranous portion of the trachea underwent avascular necrosis, while cartilage tissue maintained its viability and mechanical properties. The authors attributed this finding to the lower metabolic requirements of cartilage tissue and, for their subsequent human cases, removed the posterior membrane prior to heterotopic implantation in the left forearm [7]. Moreover, they observed that the process of revascularization could be fostered by making partial incisions into the intercartilaginous ligaments to allow for ingrowth of the recipient’s vessels into the donor’s submucosal layer or by also covering the luminal surface of the graft with recipient’s tissue. A well-vascularized submucosal layer was necessary for the survival of recipient-derived epithelial cells after withdrawal of immunosuppression [14].

The issue of revascularization of TE constructs has not been equally well studied, but it can be presumed that the formation of a recipient-derived vascular network is vital to the survival of the graft and, in particular, that of their cellular components.

Heterotopic revascularization in humans has been previously tested by wrapping the tracheal transplant in three different tissues, (i.e., the omentum, the soft tissues of the forearm, and the sterno-cleido-mastoid muscle). Beyond the various heterotopic revascularization sites, all have proven to be effective; for this reason, this option seems to be the most promising to address the problem linked to the complexity of the tracheal vasculature network and for TE tracheal substitutes.

Therefore, is likely that a similar timing applies also to a TE construct. A two-step approach involving an initial period of heterotopic implantation to allow for revascularization, followed by orthotopic implantation, has been seldom adopted in TE studies [74,75]. In these studies, however, different choices were made concerning the duration of the revascularization period (from 1 to 8 weeks), the recipient site of heterotopic implantation (sternocleidomastoid muscle or omentum), and whether the vascularized tissue was left attached to the TE construct at the moment of orthotopic implantation or not [74,75]. In this regard, Kim et al. observed better results in their in vivo canine model when the omentum, which was used as a wrap for the tracheal scaffold, could be brought to the neck together with the scaffold without separating it from its vascular connections at the time of orthotopic implantation [75].

A relatively simpler approach consists of wrapping the TE construct with a recipient’s vascularized tissue directly at the moment of orthotopic implantation. This approach was chosen by Elliot et al. in their first human case of tracheal replacement with a stem cell-seeded graft [76]. The patient was a 12-year-old boy suffering from a long-segment congenital tracheal stenosis, who was treated with transplantation of a 7 cm bioengineered tracheal allograft. The authors transposed the omentum between the heart and the trachea in order to minimize the risk of fistula and increase the vascularity of the graft [76]. Although this single-step procedure represents an appealing option, the risk of graft ischemia and loss of rigidity before revascularization occurs represents a serious concern.

Lastly, a recent report by Genden et al. demonstrated that human tracheal transplantation may succeed if the trachea is procured and implanted as a vascularized composite allograft, with distinct vascular elements that are anastomosed with those of the recipient in the same way that other solid organs (i.e., the heart, liver, and kidney) are transplanted daily [6]. To the best of our knowledge, such an approach is currently unparalleled in the field of tracheal tissue engineering. To date, fabricating and subsequent seeding with appropriate cell populations of a vascularized composite tracheal scaffold seems a futuristic concept. However, studies on whole-organ decellularization and subsequent cell repopulation are currently underway, with the aim of recreating fully functional organs such as lungs with their vasculature and bronchial tree [77]. Hopefully, a similar approach will also be applied to the field of tracheal tissue engineering.

## 9. In Vivo Animal Models

Animal models are essential to research on TE. In fact, biocompatibility assays of scaffolds with single-cell populations, such as MSCs, to test cell attachment, growth, and differentiation properties of the scaffold [78,79] or with lymphocytes to assess immunogenicity [16] are important steps in a translational pathway; however, they provide very limited information as to what could eventually happen when the TE construct is exposed to the recipient’s body environment.

Heterotopic transplantation studies provide additional information on the processes of graft revascularization and interaction with the host’s immune system and the recipient’s cell repopulation; however, heterotopic implantation sites such as the sternocleidomastoid muscle [16,49] or a subcutaneous pocket [64,79,80] do not entirely reproduce the environment of the human airway.

Orthotopic transplantation, on the other hand, is the only procedure that reliably allows for evaluation of the biocompatibility of the implanted airway graft and the capability of maintaining a patent lumen and avoiding ominous complications like infection, ischemia, wound contraction, or granulation tissue formation. In this regard, the choice of the animal model is fundamental. In fact, small animal models (rat, rabbit, or guinea pig) can aid in basic immunological graft characterization, but large animal models with comparable size and anatomy to the human trachea (dog, pig, sheep, or goat) should be used to test the viability of TE tracheal grafts [81]. Moreover, the size of the implanted airway segment and the defect type (window-like or circumferential) deserve consideration. Window-like defects are still important models for clinical translation and can be used in studies that aim at creating a non-circumferential, limited-size airway substitute. Generally, the window defect is created in the anterior tracheal surface, which mimics the clinical scenario of large post-tracheostomy tracheal defects. Intraoperative management of ventilation is relatively simple, since in most cases, the orotracheal tube can be left in place during the whole procedure (Figure 1 and Figure 2). On the other hand, circumferential tracheal defects are characterized by higher technical complexity, since cross-field intubation is required at the moment of tracheal transection. Furthermore, early animal studies have demonstrated that replacements of circumferential tracheal defects are more complicated to repair, and the risk of animal death is higher than with window-like defects [73]. To date, few studies of circumferential orthotopic tracheal replacement have been conducted on large animal models, with mixed results [17,53,75,82]. These studies provide us with the most valuable insights as to the current state of research in TE tracheal transplantation.

Lastly, as mentioned previously, attempts at transplantation of a TE airway in humans have been reported [56,76,83]. Two cases were reported by Elliot and coworkers in the setting of a compassionate treatment due to a medical emergency [56,76].

The first patient was a 10-year-old child with long-segment congenital tracheal stenosis and pulmonary sling for which he received several previous metal stents. At the age of 10, the clinical condition of the patient worsened, and an erosion of tracheal stents, creating a new aorto-tracheal fistula, was diagnosed. Fortunately, bleeding stopped spontaneously, which provided to the authors the needed time to plan, as a salvage option, a transplant with a decellularized tracheal scaffold of 7 cm in length. The TE tracheal substitute was previously seeded with MSCs from recipient bone marrow and with epithelial cells obtained directly from the tracheal allograft. The surgery was led under cardiopulmonary bypass with progressive cooling to 18 °C; as written before, the transplant was wrapped with the omentum, which was mobilized and interposed between the neo trachea and heart to reduce the possibility of future fistulae. The postoperative course was uneventful, and the patient was extubated on postoperative day 26. Despite the need to place several stents to maintain the airway’s patency due to malacia of the replaced tracheal tract, 24 months after surgery, the patient was healthy and went back to school [76].

The second patient was a 15-year-old girl born with a single left lung and long-segment congenital tracheal stenosis. For this condition, she underwent multiple surgeries, including one tracheoplasty with a lateral costal cartilage graft repair at 2 months of age, pericardial patch tracheoplasty due to recurrent stenosis following failed serial balloon dilatations, and a permanent tracheostomy. At the age of 15, the patient suffered from acute respiratory failure, with successful resuscitation. In the absence of a satisfactory conventional reconstructive option and in light of the previous case, a TE construct was offered to the child and her family. Thus, the patient received a decellularized scaffold seeded by MSCs and epithelial cells. The MSCs were harvested from a bone marrow aspirate prior to TE tracheal substitution, while respiratory epithelial cells were obtained from the nasal mucosa. Cells were seeded onto the scaffold in a bioreactor before transplantation. Unfortunately, despite an initial regular postoperative course, on postoperative day 15, an intrathoracic hemorrhage occurred, causing the death of the patient [56].

Other cases reported by the group of Macchiarini et al. [83] have raised serious concerns of scientific misconduct in the international community [84,85] and will not be discussed further here.

It is the authors’ opinion that tracheal TE is still at too early a stage for clinical translation. Several aspects, such as the ideal scaffold, cell seeding and culturing strategies, and graft vascularization strategies, still need to be clarified. Moreover, consistent and reproducible results in large animal model need to be obtained before this treatment option can be attempted with reasonable safety on humans.

## 10. Conclusions

Tracheal tissue engineering is a fascinating field of research that aims to satisfy a currently unmet clinical need. Patients affected by long-segment tracheal stenosis currently have no other option than long-term tracheal stenting, with a significant decrease in quality of life. Moreover, endotracheal stents or T tubes present significant drawbacks, such as the need to be changed regularly, the possibility of migration, granulation tissue formation, stenosis, hemorrhage, and fistula.

Compared to other treatment options, such as tracheal transplantation or stented aortic matrices, tracheal TE aims at creating a fully functional airway replacement, with no need for immunosuppression or stenting.

There is a vast body of experimental studies on tracheal TE. Both natural and synthetic scaffolds show potential for airway replacement, each with potential advantages and drawbacks. Mesenchymal stem cells, chondrocytes, and adult epithelial cells are the most commonly used cell populations used for seeding of the scaffold, and several studies suggest that co-seeding strategies could lead to more efficient scaffold repopulation. Bioreactors and administration of growth factors could further improve the efficiency of cell seeding and expansion. To date, the mechanism with which stem cells and other adult cell types contribute to improve the biocompatibility of the scaffold are not entirely clear, as they may comprise a combination of permanent cell engraftment and the release of trophic factors or immunomodulatory properties. Revascularization of TE grafts is still an open problem that needs to be considered if prolonged survival of the seeded cells on the graft is desired. Orthotopic TE airway transplantation in large animal models is a necessary step in order to obtain reliable results for clinical translation.

Overall, the state of research in the field of tracheal tissue engineering does not seem to be mature enough for the transplantation of a bioengineered airway to be attempted on humans beyond the setting of a compassionate treatment option.

## Figures and Tables

**Figure 1 bioengineering-11-00198-f001:**
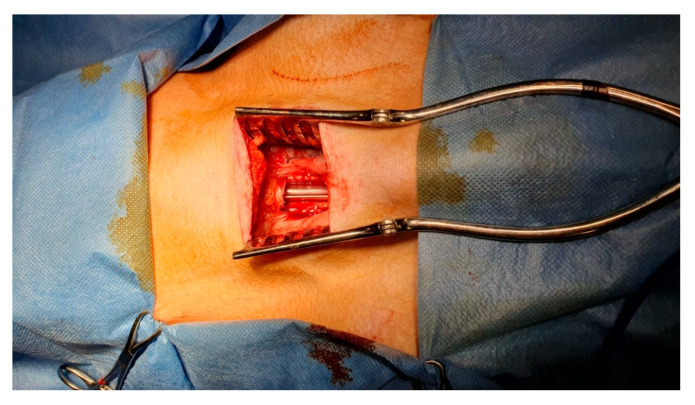
Example of a window-like tracheal defect on a large animal model (pig). Intraoperative ventilation can easily be maintained by leaving the orotracheal tube in place.

**Figure 2 bioengineering-11-00198-f002:**
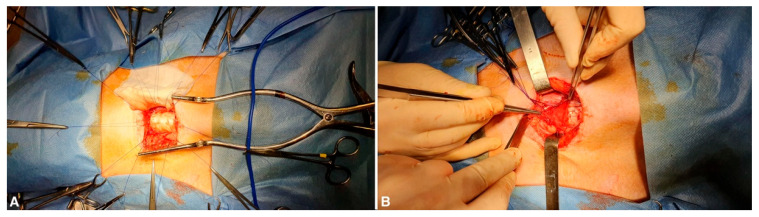
(**A**) The defect is repaired with a tracheal substitute (in this case, a decellularized tracheal segment). (**B**) A regional muscle flap (pre-tracheal muscle) is positioned over the tracheal patch in order to provide a source of vascularization.

**Table 1 bioengineering-11-00198-t001:** Main variables involved in the creation of a tissue-engineered tracheal scaffold.

Scaffold-Related Variables
Scaffold type
○ Natural ○ Synthetic ○ Hybrid (natural and synthetic layers)
Manufacturing methods (synthetic scaffolds)
○ 3D Printing ○ Bioprinting ○ Molding ○ Electrospinning ○ Salt leaching
**Cell-Related Variables**
Cell types for scaffold seeding
○ Differentiated cells ■ Chondrocytes ■ Tracheal epithelial cells ○ Stem cells ■ Adult mesenchymal stem cells ■ Embryonic stem cells ■ Amniotic fluid-derived stem cells ■ Induced pluripotent stem cells
Density of seeded cells
Seeding methods
○ Static methods (incubation) ○ Bioreactors
Growth factors
**Vascularization**
Vascular supply ○ Heterotopic revascularization (two-step approach) ○ Wrapping in vascularized tissue flap (single-step approach) ○ Vascularized composite allograft transplantation

## Data Availability

Not applicable.

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
