# Peer review of "Tracheal Tissue Engineering: Principles and State of the Art"

_bioengineering, 2024, doi:10.3390/bioengineering11020198_

Round 1

Reviewer 1 Report

Comments and Suggestions for Authors

I approached this manuscript with only a basic knowledge of tracheal biology and a general interest in tissue engineering. My immediate questions therefore focussed on what is unusual about the trachea and what particular challenges does the trachea present? Whilst the manuscript contains what seems to be a comprehensive summary of the literature I found it provided insufficient context to really answer my questions. Some specific points may clarify my concerns:

i. Why do simple acellular synthetic constructs not provide a suitable replacement?

ii. What physiology of the trachea is important? It sounds as if the epithelial cells are important in maintaining a patent airway? Are the cartilage cells important only in maintaining the collagenous matrix? Is the aim of tissue engineering to replicate the cellular population of the native tissue? Much of the discussion of stem cell types to be used reproduces many of the familiar arguments rather than indicating the particular challenges in the trachea.

iii. Similar questions relate to the biomechanics of the trachea. How important are these and, if so, this is presumably a major consideration in the choice of matrix to be used?

iv. The discussion of the importance of the vasculature is the sort of specialised issue I was hoping to learn about but procedures of wrapping various vascular tissue around the implant raises questions about the efficacy of nutrient deliver and metabolite removal: are these being investigated?

I could go on, but I hope these few points clarify my concerns.

Comments on the Quality of English Language

Language is generally satisfactory, minor grammatical errors and questionable choices of words.

Author Response

Please find attached the response from authors.

Reviewer 2 Report

Comments and Suggestions for Authors

The authors aim at producing a review on trachea engineering, however some essential topics (e.g., description of scaffold made by natural polymers) are missing and need to be addressed.

·  The authors should include a brief description of the anatomy of trachea.

· Why the authors included the study of Haykal e di Wood in the section 2, before the decellularization was introduce as a method to obtain a scaffold (paragraph 3)? The description of those studies should be moved in the following paragraph

· In grouping the scaffold in natural and synthetic, the authors do not take into account synthetic scaffold made by natural materials, such as gelatin, hyaluronic acid etc. This is an essential part that needs to be included in the manuscript.

· Regarding manufacturing technologies, the authors should also include other conventional manufacturing technologies, other then molding, such as freeze drying, salt leaching. Moreover, more details should be provided regarding 3D printing technologies, reporting also important tudies from literature.

·The authors should describe more examples from literature, comparing them and discussing about their advantages and disadvantages. Some figures, describing the most important studies should be included as well as a table to compare all the reported studies.

· Line 151-152: please cite some papers that describes this difference, so that the reader can have a refer them.

Author Response

(The authors gave the same response as above.)

Reviewer 3 Report

Comments and Suggestions for Authors

I reviewed the article entitled „Tracheal tissue engineering: principles and state of the art“. There are similar review articles summarizing tracheal tissue engineering in the literature such as:

·         Samat A.A. et al. Tissue Engineering for Tracheal Replacement: Strategies and Challenges. Adv Exp Med Biol. 2022, DOI: 10.1007/5584_2022_707

·         Dhasmana A. et al. Biomedical grafts for tracheal tissue repairing and regeneration“Tracheal tissue engineering: an overview”. J Tissue Eng Regen Med. 2020. DOI: 10.1002/term.3019

However, the review brings new perspectives on the topic of tracheal tissue engineering. There are discussed advencements in the development of tissue engineered tracheal substitues as well as drawbacks of this complicated structure regeneration.

I have only minor comments regarding the abbreviation of polycaprolactone (PCL) which is used in the text (p. 3 line 132, p.4 line157, p. 5 line 220).

Author Response

(The authors gave the same response as above.)

Round 2

Reviewer 2 Report

Comments and Suggestions for Authors

The authors adequately responds to almost all my queries. However, I think that introducing a couple of figures from most interesting/updated literatus studies is essential in a review article.  

Author Response

Answer:

Thank you for your comment. We decided to implement the paper with a table, which synthetizes main variables and issues connected with creation of a tissue-engineered trachea.

Changes:

Table 1, on page 4